# Extracellular Matrices and Cancer-Associated Fibroblasts: Targets for Cancer Diagnosis and Therapy?

**DOI:** 10.3390/cancers13143466

**Published:** 2021-07-11

**Authors:** Ismahane Belhabib, Sonia Zaghdoudi, Claire Lac, Corinne Bousquet, Christine Jean

**Affiliations:** Centre de Recherche en Cancérologie de Toulouse (CRCT), INSERM U1037, Université Toulouse III Paul Sabatier, ERL5294 CNRS, 31037 Toulouse, France; ismahane.belhabib@inserm.fr (I.B.); sonia.zaghdoudi@univ-tlse3.fr (S.Z.); claire.lac@inserm.fr (C.L.); corinne.bousquet@inserm.fr (C.B.)

**Keywords:** cancer-associated fibroblasts, extracellular matrix, cancer, therapeutic and diagnostic targets

## Abstract

**Simple Summary:**

Stroma modifications observed in solid cancer are now recognized as critical events for cancer progression and as potential therapeutic or diagnostic targets. The recent appreciation of multiple but complex roles of the extracellular matrix (ECM) in cancer, and of the cancer-associated fibroblast (CAF) diversity, has revolutionized the field and raised innovative but challenging questions. In this review, we summarize the latest knowledge regarding the role of the ECM in cancer progression, discuss the potential use of such stromal pro-tumoral modifications as therapeutic or diagnostic targets, and, finally, discuss benefits, disappointments, or even failures, of recently reported stroma-targeting strategies.

**Abstract:**

Solid cancer progression is dictated by neoplastic cell features and pro-tumoral crosstalks with their microenvironment. Stroma modifications, such as fibroblast activation into cancer-associated fibroblasts (CAFs) and extracellular matrix (ECM) remodeling, are now recognized as critical events for cancer progression and as potential therapeutic or diagnostic targets. The recent appreciation of the key, complex and multiple roles of the ECM in cancer and of the CAF diversity, has revolutionized the field and raised innovative but challenging questions. Here, we rapidly present CAF heterogeneity in link with their specific ECM remodeling features observed in cancer, before developing each of the impacts of such ECM modifications on tumor progression (survival, angiogenesis, pre-metastatic niche, chemoresistance, etc.), and on patient prognosis. Finally, based on preclinical studies and recent results obtained from clinical trials, we highlight key mechanisms or proteins that are, or may be, used as potential therapeutic or diagnostic targets, and we report and discuss benefits, disappointments, or even failures, of recently reported stroma-targeting strategies.

## 1. Introduction

More than a century ago, the “seed and soil” theory was proposed by Paget [1], seed being cancer cells and soil the stroma. Molecular characteristics of “seed (cancer cells)” were analyzed in depth and many oncogenes/suppressor genes have been identified and characterized. However, the “soil”, microenvironment encompassing host stromal cells (vascular cells, fibroblasts, immune/inflammatory cells, etc.), as well as non-cellular components (soluble factors, extracellular matrix, etc.) generated by cancer cells themselves and stromal cells, is still under characterization because of its structural and functional complexity. Recent development of novel molecular technologies has revealed the biomedical significance of the “soil” that influences cancer cell biological behaviors and functions, such as proliferation, invasion and metastatic processes. It is now clear that not only the soil promotes the growth of the seed, but also that the seed “educates” the soil to support its needs. Indeed, stromal cells acquire a specific biological phenotype via direct or indirect interactions with cancer cells. As an example, fibroblasts, which are the major components of the tumor microenvironment in most of solid tumors, become activated into cancer-associated fibroblasts (CAFs) under cancer cell stimulation, and, in turn, favor cancer development [2] notably via their secretion of acellular component such as extracellular matrix (ECM) and soluble factors. This abnormal ECM is a key regulator of tumor survival, progression and chemoresistance and represents the “breeding ground” of cancer cells. Globally, as the soil promotes the acquisition and maintenance of each of the cancer hallmarks, over the years several approaches have been used to target it.

In this context, this review will focus on (1) cancer-associated fibroblasts as main actors of matrix remodeling, (2) the impact of ECM modifications by CAFs on tumor progression, and (3) the microenvironment as a diagnostic or therapeutic target.

## 2. Cancer-Associated Fibroblasts: Main Actors of Matrix Remodeling

### 2.1. Cancer-Associated Fibroblasts (CAFs)

Cancer-associated fibroblasts, the major stromal cells of most solid cancers such as breast and pancreatic cancers, have been widely described as key actors in tumor progression through numerous mechanisms including their ability to secrete various exacerbated soluble and insoluble factors (such as ECM). CAFs are defined as all fibroblastic, non-neoplastic, non-vascular, non-epithelial and non-inflammatory cells, activated and found in tumors and metastatic niches [3,4]. As opposed to the physiological activation (observed during wound healing for example), the “activated” phenotype of CAFs is persistent [3].

CAFs have been proposed to be defined as a cellular state rather than a cell type [5] because CAF origin is diverse: whereas they mainly come from the activation of quiescent fibroblasts residing in the tumor host tissue, they also originate from bone marrow derived cells (BMDC), trans-differentiation of pericytes, endothelial and epithelial cells [6,7]. Although there are currently no specific markers defining completely and exclusively CAFs, vimentin, α-SMA (smooth muscle actin), FAP (Fibroblast activation protein), PDGFR-α (Platelet-derived growth factor receptor-α), PDGFR-β, FSP-1 (also known as S100A4) and PDPN (podoplanin) are markers [3,4,8].

During tumorigenesis, quiescent fibroblasts are activated in response to various stimuli such as hypoxia [9], oxidative stress [10], chemokines and cytokines and growth factors such as transforming growth factor superfamily (TGFβ) [11], platelet-derived growth factors (PDGF) [12], epidermal growth factors (EGF), fibroblast growth factors (FGF) [13] and sonic hedgehog (SHH) [14]. Such activation leads to an increase of CAF contractile capacities (with increased expression of α-SMA and vimentin), to a morphological modification (stellate shaped) [5], and to an exacerbated secretion of many factors (soluble and insoluble factors including ECM proteins). CAFs have been shown to first deposit fibronectin, generating intracellular tension involving actin cables. This creates, in the case of wound healing, a positive feedback loop that keeps the fibroblasts in an activated state in which YAP (Yes-associated protein) is translocated to the nucleus and α–SMA overexpressed [15]. Signals from the neo-synthesized ECM activate the Rho-ROCK-Myosin II signaling pathway and the incorporation of α-SMA into actin-myosin fibers leads to an increased contractility of activated fibroblasts [16,17]. These cells then generate tensile forces which, once transmitted to the matrix, trigger its remodeling at different levels. At the biochemical level, activated fibroblasts modify the matrix molecular composition by increasing the deposition of new matrix components, and by modulating the expression of matrix metalloproteinases (MMPs). Mechanically, these cells affect the physical properties of the matrix by modifying its organization and stiffness [18]. Such modifications induce the recruitment of new fibroblasts and their activation, and other components of ECM are produced thereby increasing the deposition of type I collagen, resulting in a decrease in fibronectin/collagen I ratio. While in a physiological context, when the collagen I network is crosslinked, fibronectin fibers are relaxed and fibroblasts resume their quiescence [15], in a tumor context, the ECM remodeling is continuous [19,20,21], altering the distribution of fibronectin zones by preventing the relaxation of fibronectin fibers [22].

Through these secretions, CAFs maintain their activated status, enhance their number and install a dialogue not only with tumor cells but also with the other stromal cells (endothelial cells, immune cells, for example) leading to complex and finely regulated tumor modifications. Indeed, while in the past researchers believed that CAFs had exclusively pro-tumoral functions (promoting tumor cell proliferation, survival, chemoresistance, angiogenesis [23] and immunosuppression [24]), in the past seven years, several publications have shown that CAF deletion or ECM modifications could result, depending on the context, in enhanced tumor progression [25,26].

One CAF particularity to take into account when trying to understand their role in cancer, is the recent observation, based on single cell RNA seq analysis, that CAFs are heterogeneous, in terms of morphology, functions and markers. This emergent concept of CAF subpopulations is based on several recent publications reporting the presence of CAF subgroups in PDAC (pancreatic ductal adenocarcinoma), breast carcinoma, colon carcinoma, lung adenocarcinoma and high-grade serous ovarian cancers [27,28] (Table 1). In breast cancer, Costa et al. distinguished four different CAF subpopulations (referred to as CAF-S1 to -S4), that accumulate differentially depending on breast cancer subtypes (luminal, HER2, and triple-negative). Importantly, as authors identified that the CAF-S1 subset highly contributes to immunosuppression, they suggest that patients with CAF-S1 rich tumors may benefit from specific immunotherapeutic strategy [29]. In non-small cell lung cancer, CAF subgrouping, either based on the collagen aspect or on fibroblast density, has prognosis significance [30,31,32]. In PDAC, David Tuveson’s team was the first to report the presence of diverse CAF subtypes: a CAF subpopulation with elevated expression of αSMA located immediately adjacent to neoplastic cells called “myofibroblastic CAFs” (myCAF), and another CAF subpopulation located more distantly from neoplastic cells (lacking elevated αSMA expression) which secreted IL6 and additional inflammatory mediators called “inflammatory CAFs” (iCAF) [33]. Later on, by performing single cell analysis, this same team also identified a third subpopulation named “antigen-presenting CAFs” (ApCAFs) capable of activating CD4+ T cells in an antigen-specific manner, thus with putative immune-modulatory capacity [34].

Over the years, several studies have looked for correlation between CAFs/ECM and PDAC patient prognosis. Erkan et al. showed in a cohort of 233 PDAC patients (who underwent surgical resection and received adjuvant therapy) that collagen deposition was an independent positive prognosis marker whereas the amount of α-SMA expression (CAF activity) was negatively correlated with patient survival, although statistically insignificant [35]. In this study, they defined four major patterns of collagen deposition with regard to PSC (pancreatic stellate cells), the main cell origin of CAFs in PDAC [3,36,37] activity, and they showed that the combination of high stromal activity and low collagen deposition was associated with worse prognosis, whereas the combination of high collagen deposition and low stromal activity was correlated with a better prognosis. Although these results are contradictory to the dogma that collagen-induced signals favor tumor aggressiveness [38,39], they have been confirmed in another publication. Indeed, Bever et al., using computer-aided quantitative method to correlate patient survival with stroma density and activity in pancreatic cancer, observed that high stromal density (ratio of the stroma area to total tumor area), but not stroma activity (measured by α-SMA expression), was significantly associated with longer disease-free survival (DFS) and overall survival (OS) in a cohort of 66 PDAC patients (who underwent pancreaticoduodenectomy and received adjuvant therapy) [40]. This year, R. Kalluri’s group reported, by using a dual-recombinase (integrating the capacity to manipulate genes using both the Cre-loxP and Flp-FRT) genetic mouse model of spontaneous PDAC to delete type 1 collagen specifically in myofibroblasts, that reducing Col1 total stromal content accelerates PanINs (pancreatic intraepithelial neoplasia) and PDAC emergence and decreases mouse overall survival by the establishment of an immunosuppressive microenvironment (recruitment of myeloid-derived suppressor cells) [41]. In contrary, Weaver’s group observed no significant association between the levels of fibrillary collagens and patient survival. The authors demonstrated, using second harmonic generation (SHG) microscopy, that the diameter of the collagen fibers adjacent to the pancreatic lesions was significantly thicker in PDAC patients with the shortest survival suggesting that collagen thickness is indicative of poor prognosis [42]. Following these findings, another group showed, using a SHG-based quantitative approach, that PDAC patients with high collagen alignment (induced by collagen crosslinking) had significantly reduced overall survival compared to patients with low alignment [43]. Then, according to these publications, collagen quality (thickness, alignment) rather than quantity is predictive of poor prognosis; therefore, not only CAF activation is important but also ECM modifications. This concept is supported by Moffitt’s studies that defined, based on PDAC virtual microdissection, “normal” and “activated” stromal subtypes with prognostic and biological relevance; “activated” subtype, corresponding to a stroma encompassing activated CAFs and highly remodeled ECM, has the worse prognosis [44].

**Table 1 cancers-13-03466-t001:** CAF subgrouping in breast, ovarian, pancreatic, lung and colorectal cancers associated with identified secretions, main tumoral characteristics and markers. Italic refers to genes. Abbreviations: Acta2: Actin Alpha 2, Smooth Muscle; C7: Complement C7; CAV: Caveolin; CCL: C-C Motif Chemokine Ligand; CD74: Cluster of Differentiation 74; Cdh11: Cadherin 11; Clec3b: C-Type Lectin Domain Family 3 Member B; CMH: major histocompatibility complex; Col: collagen; CXCL: chemokine (C-X-C motif) ligand; ENG: Endoglin; FAP: Fibroblast Activation Protein Alpha; FSP1: Fibroblast-Specific Protein-1 = S100A4: S100 Calcium Binding Protein A4; Gas6: Growth Arrest Specific 6; Gpm6a: Glycoprotein M6A; Gsn: Gelsolin; H2-Ab1: histocompatibility 2, class II antigen A, beta 1; HAS1: Hyaluronan Synthase 1; HLA-DRA: Major Histocompatibility Complex, Class II, DR Alpha; Igf: Insulin Like Growth Factor; Irf5: Interferon regulatory factor 5; ITGA11: Integrin Subunit Alpha 11; Lgals7: Galectin 7; LRRC15: Leucine Rich Repeat Containing 15; Lrrn4: Leucine Rich Repeat Neuronal 4; Ly6c1: Lymphocyte antigen 6C1 precursor; Msln: Mesothelin; MYH11: myosin heavy chain 11; MYL9: myosin light chain 9; Nkain4: Sodium/Potassium Transporting ATPase Interacting 4; PDGFR: Platelet Derived Growth Factor Receptor Alpha; PDPN: Podoplanin; POSTN: Periostin; Ptn: Pleiotrophin; Saa3: Serum Amyloid A3Slpi; Tagln: Transgelin; TGFb: Transforming Growth Factor Beta 1; Thy1: Thy-1 Cell Surface Antigen; TIMP1: Tissue Inhibitor of Metalloproteinase 1; Tnc: Tenascin C.

Cancer	CAF Subpopulations	Secretion	Main Characteristics	Markers/Key Genes
Breast cancer [29,45] and high-grade serous ovarian cancers [28]	CAF-S1	CXCL12, CCL2, CCL11, CXCL14 [28,29,45]	- Attract CD4^+^CD25^+^ T lymphocytes, promote their differentiation into Tregs and subsequent pro-tumoral functions [29] - Enhance cancer cell migration [45] - Initiate an epithelial-to-mesenchymal transition (EMT) [45]	CD29^Med^FAP^Hi^ FSP1^Low-Hi^αSMA^Hi^ PDGFRβ^Med-Hi^ CAV1^Low^ [28,29]
	CAF-S2	ND	Inactivated CAF [45]	CD29^Low^FAP^Neg^ FSP1^Neg-Low^αSMA^Neg^ PDGFRβ^Neg^ CAV1^Neg^ [28,29]
	CAF-S3	ND	Inactivated CAF [45]	CD29^Med^FAP^Neg^ FSP1^Med-Hi^αSMA^Neg-Low^ PDGFRβ^Med^ CAV1^Neg-Low^ [28,29]
	CAF-S4	CCL2, CCL11, CXCL12, CXCL13, CXCL14 [28,29,45]	Induce cancer cell invasion via NOTCH signaling [45]	CD29^Hi^FAP^Neg^ FSP1^Low-Med^αSMA^Hi^ PDGFRβ^Low-Med^ CAV1^Neg-Low^ [28,29]
PDAC [8,33,34,46,47]	Myofibroblastic CAFs (myCAFs) [33,34]	ECM proteins	Anti-tumor, contractile, stroma-remodeling	FAP^+^ αSMA^high^ IL-6^low^ *Tnc*, *Tgfb1*, *Thy1*, *Tagln*, *Col12a1*, *Pdgfrb*
	Inflammatory CAFs (iCAFs) [33,34]	IL-6, IL-11, LIF *IL-8, CXCL1-2-12, CXCL2, CCL2*	Pro-tumor, secrete cytokines and chemokines involved in cancer progression	αSMA^low^ IL-6^high^ *Clec3b*, *Col14a1*, *Gsn*, *Ly6c1*, *Cxcl12*
	Antigen-presenting CAFs (ApCAFs) [34]	ND	Present antigen to T cells	*CD74*, *Saa3*, *Slpi*, *H2-Ab1, Nkain4*, *Irf5*, *CMH class II*
	FB1 = iCAF like [46]	*Il-6, CXCL12, CCL2, CCL7*	Secretory phenotype	*Cxcl14, Ptn*, and genes mediating insulin-like growth factor signaling (*Igf1*, *Igfbp7*, *Igfbp4*), *Pdgfr*α
	FB3 = myCAF like [46]		Contractile phenotype	mesothelial markers (*Lrrn4*, *Gpm6a*, *Nkain4*, *Lgals7*, and *Msln*); fibroblast markers (*Cav1*, *Cdh11*, and *Gas6*), *Acta2* and *Tagln*, *MHC-II–associated genes*
	CAF-c1 [47]	Collagen I, SPARC, ECM proteins	Early tumors	CD74^+^*/HLA-DRA^lo^/Col1a1^+^/Col3a1^+^/TIMP1^+^/FAP^+^*, *C7^+^/ENG^+^*
	CAF-c2 = IL1-CAF [47]	Il1	Established tumors	*HAS1^+^/CXCL1^+^/CCL2^+^/FAP^+^/CD74^hi^/HLA-DRA^+^*
	CAF-c0 = TGFβ-CAF [47]	TGFβ	Established tumors	LRRC15^+^*/TAGLN^+^/Col11a^+^/ACTA2^+^/FAP^+^/CD74^hi^/HLA-DRA^+^*
	Subtype A [8]	*ECM proteins*	Associated with poor/intermediate prognosis	POSTN^high^/MYH11^low^/PDPN^low^/αSMA ^low^/PDGFRα/Vimentin^low^
	Subtype B [8]	*ECM proteins*	Associated with intermediate prognosis and with cancer cell protection against gemcitabine	MYH11^high^/POSTN^low/high^/PDGFRα/αSMA^high^/Vimentin^high^
	Subtype C [8]	*Inflammatory mediators and ECM proteins*	Associated with “good” prognostic but with cancer cell protection against gemcitabine	PDPN^high^/POSTN^low-high^/PDGFRα
	Subtype D [8]	*ECM proteins*	Associated with bad prognosis and with cancer cell protection against gemcitabine	αSMA^high^/Vimentin^high^
Non-small cell lung carcinoma [32]	High desmoplastic CAFs	ND	Enhance collagen matrix remodeling, invasion and tumor growth	αSMA^+^ITGA11^+^
	Low desmoplastic CAFs	ND	Pro-tumoral functions limited compared to HD-CAFs	αSMA^+^ITGA11^+^
Colorectal cancer [48]	PDPN^+^ CAFs	ND	Associated with prolonged disease-free survival	PDPN^+^
	PDPN^−^/α-SMA^high^ CAFs	ND	Associated with aggressive tumors	PDPN^−^/α-SMA^high^
	PDPN^−^/S100A4^high^ CAFs	ND	Associated with tumor budding and lymphovascular invasion	PDPN^−^/S100A4^high^
Melanoma [49]	S1 CAFs	*CXCL12*, *CSF1*, *CCL8*	Regulate immune cell recruitment	PDPN^high^/PDGFRα^high^/CD34^high^
	S2 CAFs	*ECM proteins*	Drive desmoplastic reaction	PDPN^high^/PDGFRα^high^/CD34^low^
	S3 CAFs	ND	Regulate actin cytoskeleton and contractility	Acta2^high^/CD34^low^

### 2.2. Extracellular Matrix (ECM)

One of the major features of CAFs is their ability to produce large amounts of ECM proteins, such as collagens, glycoproteins and proteoglycans [50]. The development of new technologies, mainly the “matrisome” approach which is based on mass spectrometry analysis of in vivo samples and enables to characterize the ECM biochemical composition, have revolutionized our understanding of tumoral ECM components and roles [51,52]. ECM is a complex scaffold composed of hundreds of proteins that provide anchoring and support to environmental cells under physiological and pathological conditions [51]. Major ECM structural components are collagens, proteoglycans, and hyaluronic acid that provide supportive framework within which other ECM components (such as laminin or fibronectin for example) and cells interact [50,53]. In addition to this architectural role, ECM proteins provide signals that cells interpret and transduce via cell surface receptors such as integrins. These signals, named mechanotransduction, activate cellular pathways that impact cellular functions such as proliferation, survival, morphology, adhesion and motility [54,55,56]. ECM represents also a growth factor reservoir as matrix proteins can sequester them and modify their signaling properties [57]. ECM modifications (biochemical composition, mechanical properties, integrity) are often observed in diseases such as fibrosis, cardiovascular or musculoskeletal diseases [58,59,60] and cancer [61,62]. It has recently become evident that ECM has biomechanical and physical properties that impact all cancer hallmarks, including the cellular processes that contribute to cancer initiation, progression, spreading [20,63] and metastatic niche formation [64,65]. Moreover, the extraordinarily dense fibrotic stroma, found in PDAC (PDAC fibrotic area accounts for up to 90% of the tumor area [66]) and breast cancer, impedes tumor perfusion and delivery of anticancer drugs [67]. ECM modifications within tumor (quantity, stiffness, etc.) have been shown to correlate with more aggressive tumors and worse prognosis for the patient [68,69,70,71].

Collagens are by far the most abundant and best characterized ECM components. Collagen I is responsible for the majority of the desmoplastic reaction [72,73,74], and high levels of its deposition have been associated with reduced survival in PDAC patients [38]. In breast cancer, as in PDAC, accumulation of fibrillary collagens I, III, and V occurs [39,75,76] and increased level of Col1a1 or Col3 is associated with a metastasis status [76] and correlates with shorter survival [77]. Mechanistically, fibrillary collagens regulate tumor cell functions via the integrin activation promoting tumor cell proliferation, migration and preventing apoptosis [78]. In contrast, type IV collagen, which is mainly present in the basement membrane (BM) that underlies epithelium and endothelial cells, is decreased in both PDAC and breast cancer [39]. In PDAC mouse model (KTC: *Tgfbr2*^flox/wt^; *Kras*^LSL-G12D/+^*Tgfbr2*^flox/wt^*Ptf1a*-Cre) [79], as in human PDAC tumors [80], not only Col4 is decreased but also BM proteins in general. This BM destruction facilitates invasion and metastasis in many cancers [63,81]. Interestingly, the tumoral ECM remodeling, involving protease-mediated ECM cleavage, generates ECM fragments, named matrikines or matricryptins, capable of influencing tumor progression and dissemination [60]. Indeed, increased level of MMP-mediated degradation of type I, II, III, and IV collagens release C-terminal collagen domains named C1M, C3M C4M and C4M12a1, respectively, that are often found in PDAC patient serum and are associated with significantly shorter survival [82,83].

Glycoproteins are the second main ECM subgroup deregulated during cancer that encompasses fibronectin, laminins and many other proteins. Fibronectin (FN) is found to be overexpressed in several cancers and reported to participate in several steps of tumorigenesis including growth, invasion, and metastasis. When analyzed as a potential prognosis factor for cancer patients, FN’s role in cancer progression appears to be complex, as FN deposited in tumor microenvironment (TME) or FN tumor cell endogenous expression have opposite correlations with patient prognosis [84,85]. A better understanding of such paradoxical role of FN in tumorigenesis is of high interest and has been recently extensively well reviewed by Tsung-Cheng Lin et al. [85]. In fibrotic solid tumors, FN expression is associated with poor clinical outcome [39], tumor aggressiveness [86], and participates in the resistance to radiotherapy via the FN-specific α_5_β_1_-integrin pathway [87]. Other glycoproteins such as periostin and galectin-1 are upregulated in PDAC [88,89] and their expressions are negatively correlated to patient survival [88,89,90]. Numbers of glycoproteins are found to be enhanced in cancer patient serum and used as diagnostic and prognostic biomarkers: for example, CA125 and CA19-9 are two glycoproteins used as ovarian cancer and pancreatic cancer biomarkers, respectively [91].

Finally, proteoglycan expression is also modified during tumorigenesis and one major example is the hyaluronan (HA). HA is a negatively charged glycosaminoglycan found to be highly accumulated in solid cancers [38,92,93]. HA expression within tumor, through a mechanism involving its high hydrophobic properties, enhances interstitial tumor pressure, and its accumulation correlates with poor prognosis [38] and metastasis [94].

## 3. Impact of ECM Modifications Induced by CAFs on Tumor Progression

Considering that fibroblast activation and ECM modifications observed in tumors are often associated with poor patient outcome, strategies interfering with the pro-tumoral role of CAFs and ECM remodeling appear as promising approaches for solid tumor treatment [95,96]. In the past decade, genetic and pharmacologic strategies to deplete CAFs leading to ECM modification were carried out in genetically engineered mouse models. Such preclinical studies and early phase clinical trials in PDAC patients have reported promising efficacy results. For example, targeting Sonic Hedgehog (shh) signaling pathway leads to a depletion of desmoplastic stroma (decrease in Collagen I content and in proliferation of αSMA positive stroma myofibroblasts) associated with an increase in vessel density, altogether resulting in enhanced intratumoral chemotherapy penetration and prolonged mice survival [97]. Unfortunately, once tested on large PDAC metastatic patient cohort in a phase I- II clinical trial (NCT01130142), the IPI 926 (Hedgehog pathway inhibitor), a well-tolerated component [98] in combination with gemcitabine, shortened patient survival mainly due to increased tumor angiogenesis [99].

This disappointing result was the first to reveal the complexity of stroma-targeting approaches. As for CAF targeting, more recent publications have reported that targeting specific ECM (HA for example) may have unexpected and dramatic effects on patient survival highlighting the importance of acquiring a precise understanding of each ECM-induced tumor function in order to develop rationalized ECM-targeting strategies and to anticipate possible impacts of such targeted strategy on all of the tumor hallmarks.

In this part, we will delineate which, and how, ECM modifications influence tumorigenesis steps (Figure 1).

### 3.1. Impact on Cancer Cell Proliferation and Survival

Cancer cell proliferation is an important part of cancer development and progression. Cancer cell proliferation is stimulated by the activation of many signal transduction pathways triggered by both intrinsic mutational and epigenetic events, and stromal-derived signals. Integrin activation, induced by ECM binding, is the major regulator of ECM-induced cell proliferation (for review, see [100]).

Key ECM components involved in cancer cell proliferation are collagens [39,75,76]. For example, type I collagen, found to accumulate in many solid cancers [101], promotes β-catenin phosphorylation, dissociation from E-cadherin and translocation into the nucleus, enhancing β-catenin transcriptional activity-dependent proliferation of gastric carcinoma cells [102]. Collagen VI, also overexpressed in some solid cancers, correlates with poor breast cancer patient prognosis [103,104], and stimulates tumor growth in part by signaling through the NG2/chondroitin sulfate proteoglycan receptor expressed on the surface of tumor cells [105]. Type V collagen which is upregulated in human breast cancer [106,107], also appears to be involved in cancer cell proliferation, as the ablation of the α3 chain of collagen V in MMTV-PyMT mouse model inhibits tumor progression by reducing glypican-1-dependent tumor cell proliferation [108].

Cancer cell proliferation and survival are also dependent on the collagen crosslinking status [109,110] mainly regulated by the crosslinking enzymes lysyl oxidases (LOXs). LOXs are copper-dependent enzymes mainly secreted by CAFs, that oxidize primary amine substrates to reactive aldehydes [111,112,113,114,115]. Levental’s group showed in vivo that LOX-mediated collagen crosslinking promotes growth and invasion of premalignant mammary organoids injected into the mammary gland [109]. Collagen crosslinking increases matrix stiffness [116] and activates the Rho/ROCK pathway to induce actomyosin-mediated cellular tension to re-establish force equilibrium [117]. This activated pathway promotes β-catenin nuclear translocation, transcriptional activation and consequent squamous carcinoma cell hyperproliferation [117].

In addition to collagen and its crosslinking, some glycoproteins are also involved in tumor growth. One of them is the fibronectin, which accumulates in several human cancers [39,87] and correlates with a shorter patient survival in colorectal cancer (CRC) [118]. Fibronectin knockdown, besides reducing cell invasion, decreases CRC SW480 cell proliferation [118]. Fibronectin promotes the activation of proliferation-related signaling pathways, *p*-ERK1/2 and cyclin D1, in glioma stem-like cells, and enhances their adhesive properties and differentiation in a concentration-dependent manner [119]. Regarding hyaluronan, a glycosaminoglycan upregulated both in the cancer cells themselves and in the surrounding stroma, and a prognosis factor for breast cancer patient survival [120], was also shown to regulate cancer cell proliferation. Indeed, hyaluronan interaction with its receptor, CD44, stimulates cancer cell growth through Rho and PI3K-AKT signaling pathways [121]. Inhibition of the hyaluronan synthase 2, an enzyme that regulates hyaluronan synthesis, inhibits breast carcinoma cell proliferation and migration in vitro and blocks metastasis in vivo, leading to prolonged animal survival [122]. All of those examples show that a global increase of ECM protein deposition and crosslinking induces biochemical and biophysical modifications that promote cancer cell proliferation.

Interestingly, some ECM proteins have opposite effects on tumor growth. A key example is the calcium-binding matricellular glycoprotein SPARC (Secreted Protein Acidic Cysteine-Rich, also named BM-40 and Osteonectin). In PDAC, SPARC is secreted by both cancer cells and CAFs [123] but very poorly expressed within pancreatic cancer cells [124]. In vitro, exogenous SPARC protein addition to PDAC cells inhibits their proliferation, while its expression downregulation increases proliferation and reduces apoptosis [125]. Surprisingly, SPARC expression in PDAC fibroblasts is correlated with a shorter overall patient survival (15 months versus 30, *p* < 0.001) and is an independent prognosis factor [126], raising the complex role of this matrix protein in PDAC progression [125].

### 3.2. Impact on Tumor Invasion

Invasion, a key step of the metastatic process, is also induced by mechanisms dependent on the expression of specific matrix proteins, ECM crosslinking/stiffness and ECM organization.

ECM remodeling in tumor involves high production of matrix proteins, ECM assembly and crosslinking, as well as ECM degradation by MMPs. This remodeling contributes to ECM stiffness, which occurs mainly through an increase in collagen deposition, crosslinking and fiber parallel reorientation [127,128]. The collagen crosslinking, dependent on LOX, generates more rigid ECM [129,130,131,132]. The increase in ECM stiffness induces in CAFs, an “outside-in” signaling via integrin engagement facilitating the formation of focal adhesions [133]. This phenomenon leads to the activation of pathways such as FAK, Rho-A or Src, acto-myosin reorganization that further increases CAF contractility [134], and maintains their activated state. These same signaling pathways activate tumor cell invasion [109,135,136], since LOX inhibition using β-aminopropionitrile (BAPN) was shown to abrogate cervical carcinoma cell invasion and migration [137].

As in CAFs, increasing ECM stiffness promotes the formation of actomyosin beams in tumor cells, that lead to increased invadopodia formation and matrix degradation dependent on their contractility [138,139]. Hence, invadopodia are actin-rich protrusions of the plasma membrane enriched in MMPs that are involved in the ECM degradation and in the formation of pores or tunnels through which cancer cells can pass [140,141]. Invadopodia were first identified in fibroblasts transformed by the oncogene v-src [142]. More recently, it has been described that CAFs are able to degrade the surrounding ECM by using a mechanism dependent on tubular organization and independent from invadopodia. Indeed, unlike invadopodia, this degradation does not require the action of Src kinase, Cdc42 or Dyn2. In contrast, inhibition of Dyn2 in fibroblasts causes a dramatic increase in stromal matrix degradation. Deterioration of ECM by CAFs requires an increase in MT1-MMP (MMP-14) cell surface expression or in MMP-2 activation [143]. This CAF-induced remodeled matrix increases tumor cell invasive capacities, illustrating how the tumor microenvironment can contribute to metastases [143]. Of course, tumor cells themselves also secrete MMPs, and ECM protein themselves can activate cellular signaling pathways resulting in MMP secretion. As an example, collagen XI*α*1, an ECM protein upregulated in several solid cancers (ovarian, breast, pancreatic, non-small cell lung cancer and glioblastoma) and correlated with poor patient prognosis, promotes ovarian cancer invasion and metastasis via the activation of the Ets-1-MMP3 pathway [144,145]. SPARC also favors cell invasion through the induction of MMP expression by CAFs and monocytes [125,146,147]. Of note, SPARC is a very interesting ECM protein that, depending on cancer type, can promote (glioblastomas, melanoma, breast cancer and prostate cancer) or inhibit (neuroblastoma, ovarian cancer, pancreatic cancer, colorectal cancer and gastric cancer) metastasis (for review see [148]). In addition to their role on MMP expression, ECM-induced integrin engagement in tumor cells also activates signaling pathways leading to tumor invasion: collagen XIII, through β1 integrin activation, enhances cancer cell invasion, migration and mammosphere formation dependent on TGF-β signaling [149]. As for collagen XIII, fibronectin and laminins also induce integrin-dependent tumor cell invasion [150,151].

Finally, in addition to ECM quantity and stiffness, ECM organization plays a critical role in tumor invasion. This is illustrated by the fact that an organizational classification of tumoral ECM, proposed on the basis of “tumor-associated collagen signatures (TACS)”, correlates with patient prognosis [152,153]. Compared to normal tissues where collagen fibers are corrugated with an isotropic orientation (TACS-1), they are found strained, anisotropic and oriented either parallel to the tumor boundary (TACS-2) in non-invasive tumors, or perpendicular to the edge of the tumor (TACS-3) in invasive tumors. While cell density impedes cell migration through ECM [154], the orientation of collagen fibers perpendicular to the tumor mass and the paracrine signals of stromal cells guiding the directional migration of tumor cells promote tumor invasion by facilitating the migration of tumor cells to the vascular system [155,156]. As for collagen, fibronectin parallel fiber organization enhances invasive velocity and directionality of pancreatic cancer cells [157].

Such matrix organizational changes are dependent on ECM composition modification [158], on integrin engagement [150], on expression of specific peptidase such as FAP (fibroblast activation protein) [157], and on actomyosin-dependent contractility of cells (CAFs and tumor cells) that allow to modify the alignment of ECM fibers [112,159,160,161,162,163]. While CAFs and tumor cells are able to generate such matrix tracks that tumor cells use to invade [164], non-activated fibroblasts, less contractile, cannot align ECM fibers and therefore do not promote directional migration of cancer cells [160,162], nor their proliferation [112]. Importantly, collagen fibers alignment correlates with tumor progression [153].

### 3.3. Impact of ECM Modifications on Angiogenesis

The vascular ECM is composed of two different compartments: the basement membrane (BM) and the interstitial ECM. In capillaries, the vascular wall is limited to the BM. This specialized macromolecular assembly of proteins separates the endothelium from the surrounding stromal tissue [165]. It forms a continuous sheet-like structure from 50 to 150 nm thick that surrounds the basal surface of the endothelial monolayer and envelops the perivascular cells, which also participate in BM synthesis and organization [166,167]. Since the capillaries lack smooth muscle cells and associated interstitial ECM, BM alone provides the structural and mechanical features that support the endothelium and vascular integrity and allows the distribution of nutrients and oxygen. The main components of vascular BM are type IV collagen, laminins, heparan sulfate proteoglycans (HSPG) and nidogen/entactin; the minor components include agrin, SPARC, fibulins, type XV and XVIII collagens [168,169,170]. Laminins and collagen IV are self-assembled into macromolecular sheets interconnected by nidogen and perlecan.

Tumor angiogenesis is dependent on BM drastic modifications [171]. Its composition and structure are modified in many ways to generate specialized and context-specific assemblages [172]. Angiogenesis is associated with degradation and reformation of BM [171]. In response to growth factors impacting endothelial cells and the presence of MMPs, BM undergoes changes in degradation and structure. This transition of mature BM into a temporary matrix promotes the proliferation and migration of vascular endothelial cells. Growth factors, such as VEGF, FGF-β and PDGF produced by tumor cells, fibroblasts and immune cells and pre-stored in BM, are released throughout its degradation by MMPs [171]. This leads to the formation of an intermediate BM, and then of a new (mature) complex matrix. BM proteins have been shown to participate in the neoformation of vessels. For example, perlecan exerts a pro-angiogenic activity mainly through the regulation of a FGF-2 signaling pathway involved in cell proliferation, motility and adhesion, and contributing to the maintenance of endothelial integrity and barrier function [173]. Moreover, this proteoglycan can bind to many growth factors and pro-angiogenic proteins, representing a reservoir of factors, that are released upon perlecan degradation by MMPs or heparanase [174]. Perlecan aberrant expression or fluctuations in its expression levels may occur during tumor progression, resulting in increased invasiveness and metastatic potential [175,176]. In contrast to perlecan which has a pro-angiogenic activity, the protein resulting from its degradation by MMPs, called endorepelin, has anti-angiogenic properties. Via its association with α2β1 integrin, endorepelin activates a signaling cascade in endothelial cells inhibiting their proliferation and migration [177].

In addition to endorepelin, other fragments of ECM protein cleavage depending on MMP activity have been reported to play an anti-angiogenic role: endostatin, a fragment of collagen XVIII, is an endogenous inhibitor of angiogenesis with anti-tumor functions [178]. However, high levels of circulating endostatin have been observed in several human cancers, such as CRC where this increase correlates positively with systemic inflammation and invasion and negatively with recruitment of mast cells and dendritic cells into the tumor depending on the anti-angiogenic role of endostatin [179]. Angiostatin is another example of endogenous inhibitor of angiogenesis produced by the action of MMP-2, -7 and -9 or -12 on precursors such as plasminogen, [180,181,182].

In addition to these ECM protein fragments, SPARC plays a key role in PDAC angiogenesis. Indeed, mouse tumors lacking SPARC show bigger tumors associated with abnormal vessels (less vessels but more permeable due to the decrease in BM and pericyte recovery around them), increased pro-tumor macrophage recruitment within primary tumor and enhanced metastasis [183]. Interestingly, the same group showed that forced expression of MMP-9 rescues the loss of angiogenesis and abrogates metastasis of pancreatic tumors triggered by the absence of host SPARC [184]. These data imply that SPARC and MMP-9 interact to regulate angiogenesis and tumor invasion in PDAC. Another matrix protein with anti-angiogenic properties is the collagen IV. Indeed, this major component of BM was shown to reduce angiogenesis and tumor growth in melanoma particularly through its NC1 domains [185,186,187] supporting the idea that these specific domains could be used as angiogenesis inhibitors.

Altogether, while in physiology BM plays the role of wall around vessels, in cancer BM becomes highly porous [188,189], thus facilitating the dissemination of tumor cells and the infiltration of pro-tumor immune cells promoting cancer progression [190,191]. Similarly, the lymphatic system can also transport tumor and immune cells. Studies have shown that the α9β1 integrin plays an important role in lymphatic vessel formation [192,193], suggesting that ECM is likely to play a role in tumor lymphangiogenesis. In conclusion, the amount of ECM, its composition, its organization as well as its cleavage proteins play a key role in tumor angiogenesis.

Finally, the extensive extracellular matrix synthesis and remodeling taking place during tumor progression generates the accumulation of biomechanical forces responsible for blood and lymphatic vessel compression, reduction of perfusion rates and increased hypoxia. While almost all matrix proteins participate in the accumulation of such forces, also named “solid stress”, collagen and HA are the main actors [194]. Collagen fibers, highly cross-linked in cancer, are remarkably stiff in tension, provide tensile strength to tissues and resist stretching, altogether contributing to growth-induced stress. On the other hand, HA, a highly negatively charged protein, provides compressive resistance because of its capacity to trap water molecules and to generate electrostatic repulsion. Water is incompressible and because water molecules cannot escape from the tumor, they resist the compressive stress developed within the tumor. HA also induces electromechanical swelling of the matrix leading to the compression of the surrounding capillaries. This electromechanical swelling matrix associated with phenomena frequently found in the tumor such as increased vascular permeability (leakage) [195] and loss of lymphatic drainage [196,197] contributes to an increase of hypoxia and interstitial fluid pressure [195,197].

### 3.4. Impact on Pre-Metastatic Niche Preparation and on Metastasis Development

Matrix remodeling not only facilitates tumor cell evasion from the primary tumor but represents also a critical event for the formation of the so-called pre-metastatic niche, a key step of the metastatic process. Indeed, the niche preparation consists of ECM remodeling and in stroma cell (BMDCs, endothelial cells and fibroblasts) recruitment and/or activation. Recent studies have shown that factors secreted by the primary tumor, such as LOX, MMPs and exosomes [198,199], initiate the formation of this niche even before the dissemination of tumor cells [199,200,201]. Similarly as for the primary tumor, in the metastatic niche, the ECM remodeling (matrix stiffening, crosslinking, etc.) favors tumor cell adhesion, migration, proliferation and survival.

In human hepatocellular carcinoma, the level of circulating LOXL2 (lysyl oxidase-like-2) is highly increased and correlates with metastasis occurrence. This LOXL2 overexpression in tumor cells is induced by hypoxia, TGF-β and SMAD4 pathways. LOXL2 secretion induces, at distance to the tumor, collagen crosslinking, ECM stiffness and durotaxis-dependent BMDC recruitment [198]. BMDCs are crucial actors in the pre-metastatic niche development [202] contributing to hepatic cancer cell invasion and colonization into the lung [115,198]. LOX expression, highly induced in breast hypoxic tumors, promotes collagen IV crosslinking in the lung, inducing CD11b+ myeloid cell recruitment. Once adhered to collagen IV, these cells secrete MMPs, leading to collagen degradation and enhanced BMDC and tumor cell recruitment [115]. It is important to note that LOX activity and expression is also highly increased in response to surgical resection of primary tumor. Indeed, Rachman-Tzemah and collaborators show that, in breast cancer, surgery-induced hypoxia promotes LOX secretion that, by enhancing fibrillary collagen crosslinking and focal adhesion signaling pathway, favors metastasis development and decreases mouse survival [203]. These results have to be carefully considered since surgery remains the most successful curative treatment for cancer and that some patients with early-stage disease who undergo surgery eventually succumb to distant metastasis.

MMPs are also key actors of the niche preparation particularly by modifying vascularization. Indeed, orthotopic grafting of melanoma or breast tumor cells compromises vessel integrity at the pre-metastatic niche site (lung) due to an upregulation of MMPs and angiopoietin. Intrapleural injection of RNAi (interference) against MMP-3/-10 and angiopoietin alleviates vessel changes, blocks myeloid cell recruitment and metastases, showing their key role in the tumor cell and myeloid cell recruitment and retention at the pro-metastatic site [204].

In addition to LOX and MMPs, tumor-derived exosomes actively participate in the metastatic niche preparation. Exosomes are small membrane vesicles (30–100 nm) that carry parental cell cargos including lipids, metabolites, proteins, nucleic acids [205,206,207,208], which enable local and systemic cellular communications [209]. Tumor cells produce high numbers of exosomes that can, at distance from the primary tumor site, remodel the ECM, induce BMDC recruitment, activate fibroblasts and modify vasculature, altogether leading to the generation of sites suitable for metastasis development [199]. As exosomes express distinct integrin expression patterns, they are differently uptaken by organ-specific cells. For example, exosomal α6β4 and α6β1 integrins, which preferentially bind to lung fibroblasts and epithelial cells, favor lung metastases, whereas exosomal αvβ5 integrins which interact with Kupffer cells rather favor hepatic metastases [199]. Uptake of PDAC-derived exosomes by Kupffer cells causes TGF-β secretion and upregulation of fibronectin production by hepatic stellate cells, via a mechanism dependent on macrophage migration inhibitory factor (MIF) expression, resulting in the development of a fibrotic microenvironment and in the recruitment of BMDC [201]. Melanoma cell exosomes are preferentially captured by sentinel lymph nodes where they induce ECM deposition and lymph node vascular proliferation, therefore favoring the recruitment of circulating melanoma cells [210]. In addition, tumor-derived exosomes can regulate MMP activity leading to ECM modification at specific sites. Indeed, glioblastoma-, breast cancer-, fibrosarcoma- and melanoma-derived exosomes contain and release MMP-2 activators, such as Hsp90 (heat shock proteins (HSPs)) and/or MMP-14, that, *in fine*, enhance cancer cell invasion by degrading collagens [211,212,213]. Altogether, tumor-derived exosomes prepare a favorable microenvironment at specific i.e., non-random, pre-metastatic niches.

### 3.5. Impact on Tumor-Associated Inflammation

ECM impacts immune cell function in tumors through both its architectural properties by acting as a physical barrier or providing invading tracks, and through its biochemical properties by trapping and releasing cytokines or inducing intracellular signaling pathways.

ECM density, organization and composition play key roles in immune cell migration and spatial distribution [57,214]. Indeed, dendritic and T cells are able to migrate along collagen I fibrils independently of integrins and adhesion molecules (amoeboid migration), whereas tumor and mesenchymal cells use a migration dependent on proteases and integrins (mesenchymal migration) in order to penetrate dense matrix [215,216,217]. In dense tumoral ECM matrix, T cells use ECM fibers as substrate for their migration and squeeze through matrix gaps, following the paths of least resistance under the guidance of chemokines secreted by cancer cells [218]. In PDAC, collagen I fibers represent guides for T cell migration that prevails on tumor cell chemokine guidance; it leads to their aberrant accumulation into pan-stromal compartment [219], a feature correlated with patient shortened survival [220]. In lung cancer, the chemokine-dependent T lymphocyte infiltration occurs in regions where fibronectin and collagen are lost, whereas it is altered in dense matrix fibers surrounding the tumor islets, leading to a preferential T cell accumulation in the stroma and to their limited intratumoral infiltration [221]. ECM properties also affect myeloid cell localization and activity in cancer. Indeed, ECM stiffness, compaction, and plasticity influence recruitment, polarization and function of macrophages [222,223]. One underlying mechanism depends on the ECM stiffening-induced tumor cell and CAF production of macrophage chemoattractants, such as CCL2 (C-C Motif Chemokine Ligand 2) and CSF1 (Colony Stimulating Factor 1) [224,225]. ECM composition, in addition to its organization and density, governs immune cell infiltration as it can enhance (e.g., fibronectin or hyaluronan) or abrogate (e.g., tenascin or versican) T cell migration and activation [226,227,228,229,230], and promote (Hyaluronan and collagen-rich ECM) or block (SPARC) pro-tumoral M2 macrophage recruitment and activation [231,232,233,234]. Thus, features of tumor ECM govern immune cell migration, localization and activation, thereby precluding T cell interaction with cancer cells and antitumoral responses [235], thus worsening patient clinical prognosis [220,236].

Moreover, ECM is a major source of molecules with immunomodulatory activities [237]. There are many examples of cytokines and growth factors secreted by cells and trapped by ECM proteins, as well as peptidic fragments produced from the matrix protease activity, which affect proliferation, migration, and differentiation of immune cells. [237]. For example, fibronectin binds to VEGF or HGF [238,239], and perlecan, abundant in ECM, binds to FGF, resulting in the sequestration of these molecules for storage [240]. The proteolytic cleavage of ECM proteins releases growth factors and contributes to localized cell proliferation and differentiation [240]. Another example is the TGFβ, which is trapped in its inactive form in ECM as it can bind to collagens or decorin [241]. Upon MMP-induced ECM remodeling, TGFβ is released, cleaved (pro-form) and thus, activated promoting immune evasion through effector T cell inhibition [242].

Finally, as many ECM proteins are composed of chemokine and cytokine-like structure domains [234], their proteolysis releases ECM fragments that can regulate many processes, including migration, adhesion and differentiation, and affect immune cell behaviors and pro-inflammatory functions [243]. Such fragments are called matrikines. For example, N-acetyl Pro-Gly-Pro (PGP) is a collagen I bioactive fragment produced by an MMP-8- and MMP-9-dependent cleavage. PGP shares a structural homology with CXCR1 (C-X-C motif chemokine Receptor 1) and CXCR2 to attract neutrophils into inflammation sites [244]. These studies show the importance of ECM fragments on immune cell migration control.

### 3.6. Impact of ECM Modifications on Chemoresistance

Many cancer cell types have been shown to use their environment interactions to acquire a drug resistance, either in the primary tumor, or in a metastasis dissemination context [245,246,247,248,249]. ECM not only participate in the establishment of a physical barrier to drug penetration (hypoxia, pH and interstitial fluid pressure (IFP)) but also, upon ECM/cell adhesion, induces chemoresistant signaling pathways within cells.

In many cases, the penetration of therapeutic agents in interstitial spaces inside and around a tumor relies on diffusion and convection under pressure and most anticancer therapeutic agents have a limited penetration in solid tumors [250]. Indeed, the increased ECM-induced IFP is one of the major factors that inhibit the penetration of macromolecules such as chemotherapy [251,252]. The electromechanical swelling matrix not only impacts the IFP but also leads to the generation of a compressive stress supported by the local growth of cancer cells [253]. This compressive stress was reported to increase PDAC chemoresistance [254]; hence, when applied to pancreatic tumor cell spheroids, such mechanical stress was shown to decrease cell proliferation, thereby reducing chemotherapy efficacy, such as gemcitabine [254]. Altogether, ECM-induced mechanical forces represent a physical barrier and strongly influence drug diffusion [255,256].

Chemoresistance has also been linked to the presence in tumors of specific ECM proteins, such as laminins, collagens, HA, SPARC and periostin. For example, laminin-332 was shown to be involved in maintaining the self-renewal abilities of human hepatic cancer stem cells and in their resistance to sorafenib and doxorubicin [257]; in stage III colorectal cancer patients, adjuvant chemotherapy drives no survival benefit for patients with tumors expressing high level of laminin β3 chain, while it does in the low laminin β3 chain group [258]; laminin- and fibronectin-induced chemoresistances are dependent on activation of the phosphoinositide 3-kinase-Akt pathway in lung, ovarian and breast cancers [257,259,260]; periostin, overexpressed in almost all solid cancers and a factor of poor prognosis in ovarian carcinoma, promotes cisplatin, 5-fluorouracil and Akt-dependent paclitaxel resistance [261,262,263]; SPARC regulates the fibrillary ECM deposition of TGFBI (Transforming Growth Factor Beta Induced), subsequently influencing ovarian cancer cell motility and decreasing response to paclitaxel [264]; collagen VI overexpression in ovarian tumor promotes cisplatin-resistance in vivo [265]; HA, via a cooperative engagement of CD44 and integrin αV, enhances glioblastoma resistance to alkylating chemotherapies [266].

Finally, cell adhesion to ECM through integrins also induces cancer cell chemoresistance via the induction of various survival pathways [64,267,268] such as PI3K/Akt, MAPK/p53, ERK/MAPK and Rho/ROCK. Based on their key role in chemoresistance, these signaling pathways have become a major axis of anticancer treatment [269,270,271,272].

## 4. Therapeutic Targeting of the Microenvironment

### 4.1. CAF Targeting

Based on extensive literature showing that CAFs stimulate tumorigenesis and drug resistance, different strategies have been developed. They are aimed at targeting either CAF activation or CAF/tumor cell crosstalk (Figure 2). Importantly, none of them are specific to CAFs as drug targets are also expressed (α-SMA, FAK, etc.) or secreted (TGF-β, ECM proteins, etc.) by other cells within the tumor.

The first strategy to be tested was CAFs depletion. An approach has consisted of the use of a vaccine that targets the protein FAP, a surface glycoprotein serine protease. This vaccine was shown to inhibit primary tumor growth and pulmonary metastases in colon cancer through increased CD8(+) T-cell-mediated tumor cell killing in tumor-bearing mice [273]. In addition, Reisfeld’s group showed that the FAP vaccine kills CAFs in breast carcinoma, thus increasing the intratumor uptake of chemotherapeutic drugs [274]. In PDAC models, the genetic depletion of FAP+ CAFs in combination with radiation was associated with increased CD8+ T cell infiltration, but did not improve animal survival [275]. Nevertheless, FAP+ CAFs depletion disrupted a CAF/cancer cell dialog through CXCR4/CXCL12 pathway inhibition, thereby restoring immune control of tumor growth [276]. In addition to FAP, another CAF marker, α-SMA, has been targeted. While α-SMA+ CAFs targeting was beneficial to suppress breast cancer metastases [277], it had dramatic deleterious effect in PDAC. Indeed, the genetic depletion of α-SMA-expressing cells in a genetically engineered pancreatic cancer mouse model increased tumor aggressiveness (with undifferentiated and highly metastatic tumor cells), ultimately reducing animal survival [278]. α-SMA+ cell depletion totally destabilized tumor stroma resulting in immune surveillance suppression associated with a decreased Teff/Treg ratio and a significant elevation in Ctla4 expression and demonstrated no therapeutic advantage even in combination with gemcitabine. However, α-SMA+ cell depletion associated with an immunotherapeutic treatment (anti-CTLA-4 antibody) prolonged the animal survival [278] suggesting that stratifying patients based on their fibrosis score might offer better responses to immuncheckpoint inhibitors. Another strategy has been to target the SHH pathway involved in PDAC stromal desmoplasia [14,279,280]. Pre-clinical treatment with the SHH inhibitor IPI-926, a compound presented by Olive et al. as a drug that in vivo depleted the tumor-associated stromal tissue, i.e., decreased α-SMA+ myofibroblast proliferation and profoundly modified the tumor vasculature, was responsible for enhanced intratumoral chemotherapy delivery [97]. However, clinical trials combining gemcitabine with Saridegib (a SHH inhibitor) were discontinued because of reduced patient survival compared to gemcitabine alone [25] (NCT01130142). The principal cause of such treatment failure was an enhanced tumor aggressiveness and metastasis due to a drastic stroma modification, e.g., increased intratumoral vascularization induced by the SHH inhibitor [99]. These studies demonstrated that, although CAFs are mainly pro-tumoral actors, their direct genetic deletion or indirect pharmacologic depletion can promote dramatic tumor stroma modifications leading to tumor dissemination and chemoresistance. Thus, this highlights the importance of targeting stromal cells with caution, and of taking into account that different CAF subpopulations may have fundamentally different or even opposite functions [5], in the regulation of tumor initiation and progression [281].

Instead of CAFs deletion, another idea of CAF reprogramming into non-activated or quiescent fibroblasts arose. TGFβ is one of the main factors involved in CAF activation [282]. TGFβ pathway inhibition has been evaluated in diverse cancers, demonstrating that CAF activity, proliferation and pro-fibrotic features were disrupted, resulting in cancer regression and enhanced therapeutic sensitivities. In colorectal cancer, as in metastatic urothelial cancer, the blockade of TGFβ signaling increased tumor sensitivity to anti-PD-L1 by facilitating T lymphocyte infiltration [283,284]. In PDAC and hepatocellular carcinoma, TGFβ inhibition significantly decreased tumor growth and is associated with reduced fibrosis in the tumor microenvironment [285,286]. TGFβ antagonists also reduced cancer cell invasiveness and metastasis in colorectal cancer [287,288]. Molecules targeting TGFβ signaling such as Trabedersen and Galunisertib are, in combination with other therapies, currently in clinical trials in solid cancers including rectal and pancreatic cancer (NCT02688712, NCT00844064, NCT01373164). In PDAC, the all-trans retinoic acid (ATRA), an active metabolite of vitamin A, was shown to restore quiescence of pancreatic fibroblasts (pancreatic stellate cells, PSCs), thereby reducing their ability to generate high traction forces and subsequent force-mediated ECM remodeling [289]. Such stroma modification resulted in cancer cell invasion inhibition [289], mechanistically through reduced proliferation and increased apoptosis of surrounding pancreatic cancer cells [290], and increased T cell infiltration into tumors, thereby improving animal survival [291]. The vitamin D receptor (VDR) activation (by the VDR ligand calcipotriol, a potent and nonhypercalcemic vitamin D analog) also holds promise in the stromal-reprogramming strategy, as it increases intratumoral gemcitabine delivery, reduces tumor volume and increases mouse survival compared to chemotherapy alone in PDAC [292]. In colorectal cancer, high VDR expression showed protective effects improving patient survival [293]. Thus, ATRA and vitamin D are currently evaluated in clinical trials, in association or not with chemotherapies, for many solid cancers.

The last strategies aimed at “normalizing” the stroma and currently evaluated, consist of targeting either pathways involved in the acquisition of CAF pro-tumoral features, or pro-tumoral pathways induced by CAFs on target cells. Regarding this last strategy, the pharmacologic inhibition of CAF-dependent chemokine signaling involving CXCR4, a receptor of the chemokine CXCL12, is a good example. Hence, treatment of KPC mice (a genetically engineered PDAC mouse model) with a CXCR4 inhibitor promoted T cell recruitment into the tumor, enhancing the efficacy of anti-PD-L1 agents, and drastically reducing PDAC progression [276]. As in PDAC, blocking CXCR4 axis in breast and gastric cancer has anti-tumoral effects. Indeed, it alleviated desmoplasia and improved immunotherapy treatment by enhancing T lymphocyte infiltration in breast cancer [294], and it decreased β1-integrin-dependent tumor cell invasiveness in gastric cancer [295]. CXCR4 inhibitors are currently evaluated in multiple clinical trials, alone or in combination with other drugs, for solid cancers. Regarding strategies aimed at normalizing the stroma by blocking CAF pro-tumoral features, over multiple examples, we chose to focus on somatostatin analogues and on FAK (Focal Adhesion Kinase) inhibitors. Since CAF-induced cancer progression is mainly dependent on CAF secretion (of proteins such as ECM components, and soluble growth, angiogenic and inflammatory factors), a pharmacological approach aimed at reducing CAF protein synthesis and secretion using somatostatin analogs was recently described as an original and interesting therapeutic strategy. Somatostatin is a natural neuropeptide that transduces an inhibitory signal on the PI3K/mTOR pathway, and thereby on the process of mRNA translation, and also on protein secretion [296]. Treatment of CAFs isolated either from human PDAC [297] or from human pituitary neuroendocrine tumor [298], with the somatostatin analog (SOM230) targeting the somatostatin receptor subtype 1 specifically expressed in CAFs, downregulated CAF-secreted molecules, including IL6, CSF-1 and CCL2. By inhibiting CAF secretory activity, SOM230 not only abrogated CAF-mediated tumor growth and metastasis in a murine xenografted model of pancreatic cancer [299,300], but also CAF-mediated drug resistance as revealed by the increased survival of KPC mice when co-treated with SOM230 and gemcitabine [297,300]. Importantly, SOM230 normalized the tumor stroma by reducing pancreatic stellate cell activity, tumor-associated macrophage presence, ECM deposits and angiogenesis, altogether leading to a robust decrease in tumor progression and chemoresistance [300].

In vivo pharmacologic FAK inhibitor treatment of various solid cancers resulted in stroma normalization by impacting angiogenesis, immune cell recruitment, CAF activation and ECM production, altogether leading to decreased tumor progression, aggressiveness and chemoresistance [301,302,303,304,305,306,307,308,309,310,311,312]. Importantly, this kinase, involved in normal fibroblast adhesion and migration, was recently shown to play a key role in CAF-mediated breast and pancreatic tumor progression. Hence, our work showed that a robust FAK activity increase (phosphorylation on tyrosine 397) in CAFs from PDAC patients (compared to fibroblasts from healthy pancreas) is an independent prognosis marker for disease-free and overall survival of PDAC patients. Specific FAK inactivation (kinase activity inhibition) in CAFs normalized the PDAC stroma in vivo, decreasing both fibrosis and immunosuppressive cell accumulation within primary tumors, which led to drastic decrease of tumor spread [313]. Fibroblastic FAK knockout in a breast cancer model also suppressed cancer metastasis via a mechanism involving inhibition of tumor cell migration induced by FAK-dependent CAF-secreted exosomes [314]. Previous studies reported that pharmacological FAK inhibition in immunocompetent genetic PDAC (KPC) [312] or PDA-grafted mouse models [315], decreased tumor-infiltration with immunosuppressive cells, and lengthened animal survival when the FAK inhibitor was combined with immunotherapy [312]. Based on these studies and on the deleterious role of FAK activity in both tumor and stromal (endothelial and fibroblastic) cells [310,313,316], pharmacologic FAK inhibitors are now tested in several clinical trials, in combination with chemotherapies and/or with immunotherapies (NCT02546531, NCT02758587, NCT03287271, NCT04109456). Preliminary results from a phase I study testing defactinib (FAK inhibitor) combined with pembrolizumab (humanized monoclonal antibody directed against PD-1 receptor) and with gemcitabine in PDAC patients showed that the regimen is well tolerated. Importantly, such treatment presents encouraging efficacy in patients with stable disease on front-line gemcitabine/nab-paclitaxel (maintenance cohort: over 10 patients, 10% had partial response, 60% stable disease, and 50% progressive disease) and patients who progressed on at least one line of therapy (refractory cohort: over 10 patients, 50% had stable disease, and 40% progressive disease, 1 not evaluated). PDAC paired biopsies (before and after treatment) revealed a modification of the immune components within the primary tumor with increased proliferative CD8+ lymphocytes and decreased macrophages and stromal density [317], validating the impact of the treatment on PDAC stromal normalization.

### 4.2. ECM Targeting

In addition to CAF targeting, several studies have evaluated the efficacy of targeting, either directly or indirectly, the CAF-secreted ECM, which not only promotes tumor progression but also acts as a barrier to chemotherapies [318,319].

#### 4.2.1. Direct Targeting of ECM Proteins

The first matrix protein being therapeutically targeted is the hyaluronan (HA), which plays a key role in tumor progression [318]. The use of hyaluronidase (HA degrading enzyme), in osteosarcoma and melanoma mouse models, was shown to promote ECM network degradation and to improve the penetration and distribution into tumors of macromolecules (e.g., FITC-dextran 150 kDa, and small hairpin RNA of PD-L1, respectively) [320,321]. In a high-HA prostate cancer mouse model, the PEGylated recombinant human PH20 hyaluronidase (PEGPH20) treatment induced an antitumor response in monotherapy, and also improved the efficacy of co-administrated chemotherapies (docetaxel and liposomal doxorubicin), probably by facilitating the chemotherapeutic agent accumulation into the primary tumor [322]. In pancreatic cancer, PEGPH20 was shown to decrease tumoral water and IFP, leading to tumor vasculature decompression, increased tumor vascular perfusion and enhanced drug delivery [318,319]. Clinical trials were performed based on those encouraging studies. A clinical trial (HALO 109-202, phase 2) evaluating the combination of PEGPH20 with gemcitabine and nab-paclitaxel chemotherapy as first-line therapy in patients with metastatic PDAC, shows a progression-free survival significantly improved especially in a subgroup of patients with high level of intratumoral HA [323]. Based on these promising results, a phase III study of PEGPH20 plus gemcitabine and nab-paclitaxel (compared to gemcitabine and paclitaxel) has been conducted on patients with high tumor HA content (immunohistochemistry pre-screening). Results were disappointing as the overall survival of patients within the PEGPH20 plus chemotherapy arm was 11.2 months compared to 11.5 months in the chemotherapy arm (HR 1.00, *p* = 0.97), and PFS was 7.1 months in both arms (HR 0.97) [26]. Another study combining PEGPH20 with modified FOLFIRINOX (mFOLFIRINOX) versus mFOLFIRINOX alone (NCT01959139, phase Ib/II) on 138 treatment-naive patients with metastatic PDAC, revealed a detrimental effect of the PEGPH20 leading to shorter patient overall survival [324]. One important feature to take into account in this disappointing result is that, due to increased toxicity in the PEGPH20 arm, the number of FOLFIRINOX cycles had to be reduced in this arm compared to mFOLFIRINOX alone (median of four cycles versus height). Nevertheless, four out of 55 patients in the mFOLFIRINOX plus PEGPH20 arm presented a complete response (CR) compared to zero CR in mFOLFIRINOX arm. Altogether, these disappointing results led to PEGPH20 program discontinuation. Nevertheless, better understanding the reason of such failure may allow to propose combination therapies that may benefit patients. Experts have proposed some very interesting hypotheses [325] that require further investigation.

Besides hyaluronan, collagens are also targetable. Indeed, therapies using collagen-degrading enzymes such as collagenases are currently evaluated in the context of cancer. As drugs have difficulties reaching the tumor because of an elevated IFP, the use of collagenases is thought to be a good strategy because these enzymes degrade the ECM and therefore reduce IFP. In human osteosarcoma xenografts, Eikenes’s group observed that collagenase-induced IFP reduction led to increased drug uptake (monoclonal antibodies) into the tumor [326]. Similarly, in a melanoma model, the diffusion of a 500 kDa molecule (FITC-dextran) was shown to be 40 times decreased, compared to its diffusion in water. Intratumoral injection of collagenase or cathepsin C (an enzyme degrading decorin) reversed at least partially this phenomenon [327]. However it is important to take into account that degrading the ECM may have adverse effects such as enhanced metastasis for example (for review see [328,329,330]). Indeed, many publications reported the key role of endogenous collagenases (such as MMP1, 8 and 13) in tumor invasion, leading to the idea that MMPs should be inhibited to improve patient survival [331,332]. Such strategy, consisting of targeting indirectly the ECM, is developed in the next paragraph.

#### 4.2.2. Indirect Targeting of ECM Proteins

As mentioned, the ECM-remodeling enzymes MMPs appear to be interesting therapeutic targets, since they were described to present multiple pro-tumoral roles, including in tumor spread, angiogenesis and chemoresistance [333]. As these endopeptidases are overexpressed in multiple solid cancers in correlation with poor patient prognosis [334,335], multiple clinical trials were developed to test the increasing therapeutic interest of agents targeting MMPs [332]. Despite promising results in preclinical studies, such as the in vitro and in vivo inhibitory effect on tumor cell spread of the MMP-9 inhibitor in colorectal cancer [336], clinical trials using MMP inhibitors, e.g., Tanomastat, Prinomastat, Rebimastat were unsuccessful and mostly stopped in phase III because they showed severe side effects, no improvement of patient survival and no reduction of tumor growth [333]. In order to increase safety and specificity, researchers have developed monoclonal (humanized or not) antibodies directed against specific MMPs. Andecaliximab, a fully humanized antibody against MMP-9, showed anti-tumoral effect in a colorectal cancer preclinical assay [337], and is currently under evaluation in phase II in combination with immunotherapy (NCT02864381) and phase III with chemotherapy (NCT02545504) for gastric or gastroesophageal junction (GEJ) adenocarcinoma patients. DX-2400, DX-2802 and DX-2712 are three other monoclonal antibodies directed against MMP-14, MMP-9 and MMP-12, respectively, currently evaluated in preclinical cancer models [332,338].

Besides MMPs, heparanase, another enzyme responsible for ECM degradation and remodeling, and found to be upregulated in several tumor types in association with the presence of metastases [339], was considered as an interesting target [340]. The heparanase inhibitor, SST0001, was shown to inhibit tumor cell proliferation and to decrease VEGF, HGF and MMP-9 levels in multiple myeloma [341], to abolish the increased pancreatic cancer cell invasion dependent on radiotherapy-induced heparanase level [342], and to delay sarcoma tumor growth (when administered alone), or even induce a complete regression (in 5 out of 8 mice), when administered in combination with Irinotecan [343]. This inhibitor, also called Roneparstat, was tested in a phase I clinical trial in patients with relapsed/refractory multiple myeloma (NCT01764880). Despite excellent safety and tolerability profiles, Roneparstat showed very modest efficacy [344], and its efficacy in combination with other therapies has to be tested. Necuparanib, a multi-targeting heparan sulfate mimetic (rationally engineered low-molecular-weight heparin), showed promising efficacy and safety in phase Ib evaluation, but showed no benefit (no improvement of overall survival) when combined with gemcitabine/nab-paclitaxel in untreated (first line treatment) metastatic PDAC patients as evaluated in a randomized multicenter phase II trial (NCT01621243) [345]. Another heparanase inhibitor named Muparfostat (Pi-88) is currently under clinical evaluation (NCT00130442, phase 2) in combination with Dacarbazine in patients with metastatic melanoma.

As a matrix indirect targeting, the inhibition of the lysyl oxidases, proteins responsible for collagen crosslinking and stabilization, has shown mitigated results. LOX inhibition (through BAPN) not only reduced collagen crosslinking and fibronectin assembly but also increased drug penetration and re-sensitized tumors to doxorubicin treatment in breast cancer [346], and enhanced sensitivity to radiotherapy in prostate cancer [347]. In PDAC, when given as a single drug at very early time of the disease (in KPC mice), a LOX function-blocking antibody reduced ECM crosslinking and inhibited metastases [348]. The combination between this antibody and gemcitabine prolonged tumor-free survival of KPC mice with early-stage tumors but not with locally advanced tumors (comprising a well-established and already cross-linked matrix) [348], highlighting the fact that therapeutic agents targeting ECM and ECM remodeling may have beneficial effects depending on the tumor stage. However, a recent study, by H. Jiang et al., reported that, in multiple immunocompetent experimental PDAC mouse models, anti-LOXL-2 treatment reduced fibrosis, decreased tissue tension reduction which, in turn, enhanced tumor progression (even in combination with gemcitabine) [349]. Nevertheless, a clinical trial combining a LOX inhibitor (Simtuzumab) with gemcitabine given as a first-line PDAC treatment was initiated but, unfortunately, did not improve clinical outcomes maybe due to the metastatic stage of the recruited patient [350]. As for PDAC, combining Simtuzumab with FOLFIRI (leucovorin plus irinotecan and fluorouracil) in patients with metastatic and KRAS mutated colorectal carcinoma did not improve the clinical outcome.

#### 4.2.3. Targeting of ECM-Induced Intracellular Signaling

Other approaches have been developed to target the ECM-induced intracellular signaling [56].

The first strategy consists of targeting the main cellular ECM receptor: the integrins. Different approaches such as the inhibition of integrin function or of its downstream signaling, have been developed in cancer. For example, Cilengitide, an antagonist of the αVβ3/αVβ5 integrins, inhibited cancer cell proliferation of both primary cultures and cell lines of head and neck squamous cell carcinoma [351]. Endostatin, an endogenous antagonist of angiogenesis and tumor growth, and an αVβ1 integrin inhibitor, inhibited NF-κβ-induced CXCL1 expression and hemangioendothelioma tumorigenesis [352]. Moreover, in colorectal cancer, the inhibition of collagen-α2β1 integrin binding (using E7820 inhibitor) led to the inactivation of the ECM-induced PI3K/AKT/Snail signaling pathway, and potentiated the efficacy of chemotherapies (oxaliplatin and 5-fluorouracil) [353]. Multiple clinical trials have tested the efficacy of combining chemotherapies with integrin inhibition (e.g., Abituzumab, a humanized monoclonal antibody targeting against integrin αν heterodimers, or Cilengitide, a selective inhibitor of αvβ3 and αvβ5 integrins), but none of them improved cancer patient outcomes (in glioblastoma, advanced non-small-cell lung cancer, recurrent and/or metastatic squamous cell carcinoma of the head and neck, or colorectal cancer) [354,355,356,357,358].

Matrix modifications and mechanotransduction regulate and are regulated by various proteins (including mechanosensors) and signaling pathways (such as FAK, MMPs, SerpinB2, RhoA, JAK/STAT, YAP/TAZ, CDK4 and PAK). Such proteins, which play a key role in mechano-reciprocity and cancer development, can be targeted to prevent tumor progression [42,312,359,360,361,362,363]. FAK is a key mechanotransduction regulator, controlling both outside/in and inside/out signaling pathways. This mechanosensor is activated upon ECM-induced integrin engagement dependent on ECM modifications such as increased ECM stiffness and pressure [364]. As mentioned in the preceding sections, FAK activity is increased in stromal cells in tumors [313], but also in many cancer cells, enhancing tumor cell migration/invasion [365], proliferation and chemoresistance [306,316]. FAK inhibitors are currently tested in more than 20 clinical trials for solid cancers (ovarian, pancreatic, non-small lung, lung, mesothelioma, lymphoma, melanoma, breast, etc.).

ECM modifications also activate the ROCK signaling pathway, which is a key regulator of actomyosin contractility and of cell shape, and thereby controls functions such as cell proliferation, differentiation, motility and adhesion. ROCK1 and ROCK2 expression was shown to be increased in human pancreatic tumors, in correlation with shorter patient survival [366]. ROCK2 activation in non-invasive pancreatic cancer cells promotes their invasiveness in a collagen matrix in association with an increased ECM remodeling [366]. Increased cell contractility also triggers stiffening of the ECM and of the whole tissue [367,368], and transient tissue priming by Fasudil (a ROCK inhibitor) reduced pancreatic cancer fibrosis, thus improving response to gemcitabine/Abraxane at both primary and secondary sites [369,370]. On breast cancer cells, Fasudil inhibited cell migration [371] while it led to a pro-migratory cell phenotype in colon cancer because of an over-activation of the Na_V_1.5 voltage-gated sodium channel [372]. Altogether, the use of ROCK inhibitors may be promising for anticancer purposes but should be tightly controlled and probably restricted to specific cancers.

#### 4.2.4. New Perspectives for ECM as Circulatory Tool in Solid Tumor Diagnosis

In the last decades, cancer research has focused on the identification, based on histological parameters or gene signatures, of ECM/stroma characteristics that may have diagnostic and/or prognostic significance in most of cancers [373,374,375,376,377,378]. The tumor stroma is also a potential source of new biomarkers such as molecules generated by the matrix remodeling and released in the bloodstream [82,379]. Therefore, there is now a growing interest in studying such circulating biomarker molecules.

In breast cancer patients, a positive correlation between expression profiles of specific matrix proteins in the tumor and in the blood stream (as quantified by immunohistochemical staining or ELISA, respectively) was shown and differentiation between breast cancer patients and those with a benign disease was possible using the blood markers [380]. Collagens are the main proteins of stromal origin detected in the blood. In particular, collagen IV expression (measured by enzyme immunoassay) is significantly higher in breast cancer patient sera than in healthy donors [381]. Other ECM-derived molecules have been detected in the plasma, such as the cartilage oligomeric matrix protein (COMP) and fibronectin. In particular, the combined presence of COMP, collagen XI and collagen X in the blood makes it possible to discriminate between patients with breast cancer, healthy donors and patients with benign breast disease [380].

MMP-9 level in patient serum was shown to be significantly higher in CRC patients than in healthy subjects [382,383,384,385,386]. However, MMP-9 proteolytic activity in the serum of patients with adenomas remained systematically in the same range as in healthy subjects, suggesting the interest of this biomarker in differentiating patients with precancerous lesions (adenomas) from those with CRC [384]. Another protein that enabled to distinguish between malignant and non-malignant lesions is the tissue inhibitor of type 1 metalloprotease (TIMP-1), which was shown to be significantly higher in serum from CRC patients than from patients with colorectal adenoma or healthy subjects [385]. The collagen degradation fragments, C1M, C3M and PRO-C3 (from collagen I, III and immature collagen III, respectively) were not only increased in the serum of CRC patients as compared to that of patients with adenomas, but also proved to be discriminating serum biomarkers between the metastatic IV stage and all other stages in CRC [387].

For PDAC, in view of the urgent clinical need for early diagnostic biomarkers, stromal-derived circulating molecules may be of particular interest. Among those, collagens and their degradation fragments were the most frequently identified PDAC diagnostic biomarkers. Hence, a pilot study revealed an increase of MMP-generated collagen I, III and IV fragments in the serum of PDAC patients compared to healthy subjects [82,388]. A panel of MMPs was also analyzed in the serum of PDAC patients, showing that the concentrations of MMP-1, -3, -7, -9, -10 and -12 were increased, and of MMP-2 decreased, in those patients compared to healthy subjects, and that MMP-7 and MMP-12 were discriminating markers [389,390]. A recent study demonstrated that the combination of CA19.9 (a pancreatic tumor-associated antigen used in clinic) with MMP-7 and connective tissue growth factor (CCN2) differentiates PDAC patients from healthy patients [391], whereas a panel composed of CCN2, plasminogen (PGL), fibronectin, collagen IV and CA19.9 distinguishes PDAC patients from patients with chronic pancreatitis [391]. Franklin and colleagues evaluated ECM-derived fragments (type IV collagen and endostatin/type XVIII collagen) and matricellular proteins (osteopontin and tenascin) as biomarkers of PDAC [392], and found that the levels of those four ECM fragments are elevated in the circulation at PDAC diagnosis, compared to healthy patients. In comparison, conventional cancer cell-derived tumor markers (i.e., cancer antigens CA 19.9, CA 125, CEA and TPS (tissue polypeptide specific antigen)), were also found upregulated but their level ranges were broader. The authors concluded that combining both tumor stroma-derived and cancer cell specific proteins might improve the sensitivity and specificity of a diagnosis panel test. Importantly, high levels of a panel of stromal markers (i.e., type IV collagen, endostatin/type XVIII collagen and osteopontin) were reported, when analyzed in the post-operative setting, as predictive of poor overall survival, while conventional markers were not [392]. Another group also reported a biomarker signature comprising TFPI (plasma tissue factor pathway inhibitors), tenascin C (TNC-FN III-C) and CA 19-9 that significantly improves, compared to CA 19-9 alone, early detection of PDAC (i.e., at a stage amenable to surgical resection of patients) [393]. Similarly, immunoassay dosages of TIMP1 and LRG1 (leucin rich alpha 2 glycoprotein) were reported to significantly improve, compared to CA19-9 alone, PDAC detection at an early stage [394].

In conclusion, these studies demonstrated that ECM remodeling is a promising source of new biomarkers in tumor progression. The search for new circulating tumor biomarkers beyond the cancer cell itself is a new non-invasive diagnostic and prognosis approach.

## 5. Conclusions

Cancer is the first or second leading cause of death before the age of 70 years in 112 of 183 countries. Patient survival is dependent on, of course, cancer type, but, less excepted, on its own composition. Tumor microenvironment has been shown to influence patient prognosis, but understanding the role of each of its components, in order to target the ones implicated in tumor progression and thus in patient death, is challenging. Not only CAF subpopulations have been identified and appear to have opposite roles but also ECM quantity, quality, organization and stiffness can differentially impact cancer progression.

Based on our current knowledge and on scientifically fascinating and clinically promising data, clinical trials targeting either CAFs (activation, acquisition of CAF pro-tumoral features or CAF-induced pro-tumoral pathways) or ECM (direct and indirect targeting of ECM proteins or ECM-induced intracellular signaling pathway) have been developed. Unfortunately, most of them failed and have been discontinued. However, some showed promising results, and, as opposed to strategies aiming at destroying the tumor microenvironment (PEGPH or CAF depletion), those encouraging trials are aimed at reprogramming CAFs into non-activated cells (ATRA or vitamin D), or “normalizing” the stroma by targeting signaling pathways involved in its pro-tumor features (SOM230 or FAK inhibitors) (Figure 2).

Importantly, it is now clear that ECM proteins may help with cancer patient diagnosis. Indeed, as a consequence of matrix remodeling in cancer, a high number of ECM proteins and fragments are released in blood and can be used as circulating biomarkers for clinical diagnosis. It could be particularly interesting for very aggressive tumors, often diagnosed at non-resectable advanced stage such as PDAC for example.

## Figures and Tables

**Figure 1 cancers-13-03466-f001:**
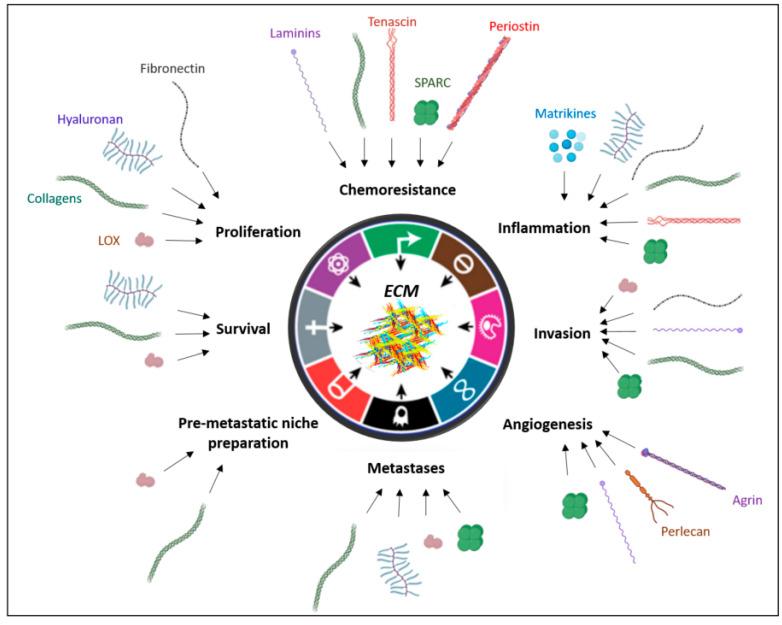
Schematic of the impact of ECM proteins on tumor functions. LOX: lysyl oxidase; SPARC: secreted protein acidic and cysteine rich.

**Figure 2 cancers-13-03466-f002:**
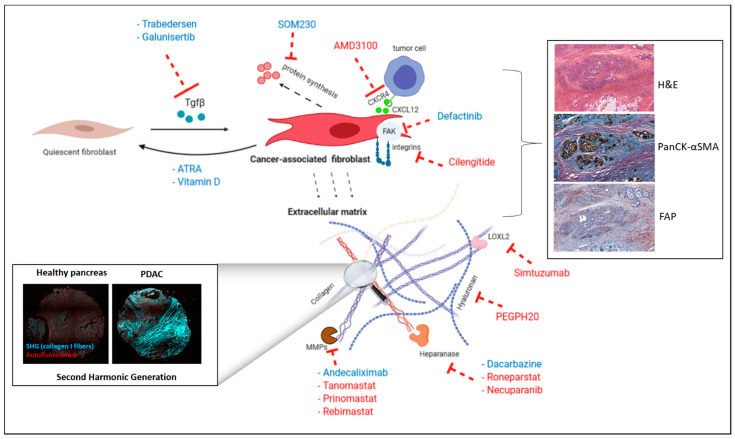
Schematic of stroma key events involved in tumor progression and under investigation in clinical trials complemented by representative images of the typical organization of ECM, CAFs and tumor cells within PDAC samples. Blue: ongoing clinical trials; red: drugs that failed in clinical trials; ATRA: all-trans retinoic acid; MMPs: matrix metalloproteinases. Representative images of H&E (allowing the visualization of ECM rich areas) and IHC on human PDAC samples showing CAFs (positive for FAP-α: fibroblast activation protein and/or αSMA: alpha smooth muscle actin), and epithelial cells (positive for PanCK: pan cytokeratin). Courtesy of Dr Jérome Cros (Department of Pathology; Hospital Bichat-Beaujon, Paris, France). Representatives images of Second harmonic generation microscopy on tissue microarray from healthy pancreas or PDAC patients showing collagen fibers.

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
