# Peer review of "Extracellular Matrices and Cancer-Associated Fibroblasts: Targets for Cancer Diagnosis and Therapy?"

_cancers, 2021, doi:10.3390/cancers13143466_

Round 1
Reviewer 1 Report
This manuscript is very well written and organized, and after revision will be a valuable contribution to the science. I agreed to review this article due to its subject of CAFs. This is a subject that is very interesting at the moment and is the subject of work that I collaborate on. I was excited to read this and that excitement continued until page 5...6...7... This is an extremely long read which diverts from its main goal somewhere along the line. This is in essence two complete conjoined reviews. The extensive review of the ECM in other cells and tissues has been done countless times over. What I was interested in was the CAF aspect....but you lost me not even half way through the paper. I would like to see a CAF specific paper as the title suggests. This paper could be cut down considerably and be a much better and targeted manuscript.
Reviewer 2 Report
The manuscript by Belhabib et al provides a review of the properties and roles of cancer-associated fibroblasts and the extracellular matrix in cancer and their potential as targets for diagnosis and therapy, focusing mainly on the extracellular matrix part. This is a timely, detailed, interesting, and well-written manuscript on a subject that is under intense investigation in recent years. My only major comment is that a figure that incorporates the information provided in section 3 would be helpful and informative.
Minor comments include:
- Lines 68-70: PDGF, EGF, and FGF are also growth factors, but appear distinct from the general characterization of growth factors (line 68).
- Line 75-77: This example is from wound healing. Is there a relevant example from cancer? If the authors want to include it, it should be clear that it refers to wound healing and fibroblast to myofibroblast activation.
- Lines 121, 243: These references are not incorporated correctly in the Reference list. Lines 866, 871-872: These references are not correctly cited with numbers.
- Lines 349 – 357: Information on the role of SPARC could be grouped together.
- Line 649: “reprogramming” instead of “reprogrammation”.
- Line 650: “factors” instead of “factor”.
- Line 658: Omit “colon and”
- Line 752: “improved” instead of “improving”.
- Line 759: the type of metastatic cancer is missing.
- Line 777: Do the authors mean “… that may benefit patients”?
- Line 829: The sentence “…, but when evaluated.” seems unfinished.
- Line 862: Do the authors mean “and resulted in hemangioendothelioma tumorigenesis inhibition”?
- Line 863-866: This sentence is not clear.
- Line 880: “for example” seems redundant.
Author Response
"Please see the attachment

Reviewer 3 Report
The review is well written and comprehensive. Very important is table 1. However, I think that it could be nice to include also other examples of tumors. Namely, the tumours from the neuroectoderm will be interesting. Figure 1 demonstrates potential therapeutic targets including therapeutics that failed in course of trials. I think that the inclusion of real histological sections demonstrating the position of CAFs and extracellular matrix molecules in tumor samples will be interesting for reader.
Reviewer 4 Report
In the review article, “Cancer associated fibroblasts and extracellular matrices: targets for cancer diagnosis and therapy?”, the authors present a thorough review on a large topic of interest. The authors briefly review 390 references as they relate to cancer associated fibroblasts, extracellular matrix and/or therapies targeting these two contributors to the tumor microenvironment.
Overall impression: The manuscript appeared well organized, and logically presented at first glance. However, the extremely broad topic ended up causing division of ideas across subtopics which is likely hard to avoid when referencing 390 manuscripts. This is expected to some degree as fibroblast secretion of collagen, for example, could logically be placed in a fibroblast subsection or an ECM subsection. However, the authors do not focus on any one (or small number) of cancer types. Nor do they focus on any particular area of signaling or category of molecules or ECM of interest. As such, the review briefly mentions multiple tumor types, multiple areas of signaling and becomes what could be organized as a bulleted list of summaries. To be clear, the authors do comprehensively review A LOT of material, but with that much material covered, it needs good organization so the reader can identify topics of interest within the 20 page review. A theme helps guide a review also, especially with such a broad range of manuscripts to help guide what is critical to maintain the theme.
Major points:
- Glioblastoma, PDAC, melanoma and colorectal cancer (to name a few that are referenced) are a broad range of cancers in a diverse set of organs. Wouldn’t we expect them to have different normal and pathologic microenvironments? What makes them different? Right now examples are mixed together without a proper introduction of the differences and proper introduction may take up too much space and the authors may consider a narrower focus. Again, this seems to be a “theme issue”.
- For example, section 4.2.2 mentions colorectal cancer, gastic cancer, GEJ while exploring clinical and preclinical assessment of MMPs and MMP inhibitors. There is no clear introduction of why one would target one MMP and not another and in what situation.
- This is not the only example. For every paragraph, the authors need to decide how the examples provided are tied together. A narrower topic with a theme would help. Right now, many paragraphs run the risk of being a bullet point list under one category heading that loosely unites them. This was more problematic in sections 3 & 4.
- Most of the molecules mentioned are familiar to me and would need to be for any reader. So, the review is targeted to people who already know some of the topic, but want a more thorough review. Otherwise, there is not enough context for a reader to know where HGF, CXCL1, SPARC, etc. fit into cancer biology other than the specific reference that is provided.
- The discussion of failed clinical trials could be simplified by leaving out details, with a brief summary and then more important discussion of where this guides the field generally and clinically. Does failure in one organ site say anything broadly or about other organ sites?
- Be careful when presenting therapeutic targeting of molecules that are not specific to the tumor microenvironment. Yes, CAFs may have aSMA, but so do vessels (throughout the body). CXCR4, FAK, etc. is often expressed on tumor cells.
Minor points
- Double check that statements contain proper references. References needed on lines 57, 99, 133 (maybe), 305
- Explain “morphological modification” and “supra-cellular structures” on lines 72 & 75.
- Is PDPN a CAF marker to be included in lines 63-66?
- Paragraph that starts on line 100 seems like it needs a new introductory sentence (since the paragraph seems to focus on CAF subtypes, which is not clear from the first sentence
- “et al” not “and al” on line 106
- Improper reference formats on line 121, 243, 871
- “Dual recombinase” on line 139 is vague.
- C1M, C3M, etc (lines 212-214) need more introduction.
- Some abbreviations are defined multiple times
- Line 228: biomarkers of what?
- Section 3.3 fist 1.5 paragraphs includes a ton of information on BM. How is this related to the topic? Is this really important, if so, how?
- Line 521: is gelatin not collagen?
- Is TGFbeta ligand in the ECM (as discussed around lines 560)?
- Galunisertib and LY2157299 are the same drug (line 659)
